# Characteristics of Kawasaki Disease Patients during the COVID-19 Pandemic in Japan: A Single-Center, Observational Study

**DOI:** 10.3390/children8100913

**Published:** 2021-10-13

**Authors:** Shoichi Shimizu, Mamoru Ayusawa, Hidetoshi Go, Kimitaka Nakazaki, Hidemasa Namiki, Yuki Kasuga, Koji Nishimura, Koji Kanezawa, Tamaki Morohashi, Ichiro Morioka

**Affiliations:** Department of Pediatrics and Child Health, Nihon University School of Medicine, Tokyo 173-8610, Japan; shimizu.shoichi@nihon-u.ac.jp (S.S.); go.hidetoshi@nihon-u.ac.jp (H.G.); nakazaki.kimitaka@nihon-u.ac.jp (K.N.); hidemasa.namiki@nihon-u.ac.jp (H.N.); kasuga.yuki@nihon-u.ac.jp (Y.K.); nishimura.koji@nihon-u.ac.jp (K.N.); kanezawa.koji33@nihon-u.ac.jp (K.K.); morohashi.tamaki@nihon-u.ac.jp (T.M.); morioka.ichiro@nihon-u.ac.jp (I.M.)

**Keywords:** coronavirus disease 2019, Kawasaki disease, Japan, severe acute respiratory syndrome coronavirus type 2, pandemic, vasculitis

## Abstract

Background: Under the Coronavirus disease 2019 (COVID-19) pandemic, manifestations in children with Kawasaki disease (KD) are different between the Western and the Eastern countries. Particularly, there has not been a report comparing a series of KD in Japan, where KD was originally discovered and has a large number of registered cases. Methods: We compared patients with KD under the period of the COVID-19 pandemic in Japan with the report from Italy during its reported period by a retrospective, cohort, observational study in a Japanese single center. Results: Thirty-two patients with typical KD were treated during the study period, while the Italian study reported 10 patients with the signs of KD. Concerning the proof of severe acute respiratory syndrome coronavirus type 2 (SARS-CoV-2) infection, none (0%) of our KD cases showed a positive result and one and no patients developed the macrophage activation syndrome (MAS) and Kawasaki disease shock syndrome (KDSS), respectively; however, eight (80%) patients in the Italian series were confirmed with SARS-CoV-2 infection. MAS and KDSS developed in six and five patients, respectively. Conclusions: Cases reported as COVID-19 pandemic-related KD in Italy showed significantly different clinical characteristics from the typical KD symptoms known in Japan. Although they show KD-like manifestations, we cannot conclude that SARS-CoV-2 has the same etiology of our ‘classic’ KD at the present stage.

## 1. Introduction

The new coronavirus (severe acute respiratory syndrome coronavirus type 2, SARS-CoV-2) outbreak, which broke out in Wuhan, China, in December 2019, has spread to many countries around the world and continues to rage. Infections caused by this new virus can range from asymptomatic to minor symptoms of the common cold and, in some patients, later progressing to respiratory failure and multiple organ damage due to severe pulmonary lesions [1]. The disease caused by this virus has been named Coronavirus disease 2019 (COVID-19) by the World Health Organization [2]. While severe cases and deaths are a serious problem in adults, the number of children developing the disease is low from the pandemic onset, and, even if infected, children have few symptoms [3]. In Japan, no deaths in children under 19 years of age have been reported so far.

Incidentally, Kawasaki disease (KD) is relatively common among Japanese children, with 17,364 cases in 2018 [4]. KD was originally discovered in Japan, and Japanese nationwide surveillance officially reports the prevalence in 2018 was as high as 359 patients/100,000 children aged 0–4 years [4]. Differences in prevalence have been observed between races, and the disease has been considered relatively rare in the Western countries [5]. However, between March and April of 2020, eight children with severe inflammatory shock were seen during the SARS-CoV-2 epidemic, first reported in the United Kingdom [6]. In the outbreak of SARS-CoV-2 infection in Bergamo province, Italy [7], which we compared in the present study, KD was found in a series of 10 cases with symptoms seen in succession. In addition, from the United States and France, there has been a rapid increase in the number of critically ill patients requiring intensive care among young patients confirmed to have been infected with each outbreak. They have called those cases multisystem inflammatory syndrome in children (MIS-C) [8,9,10,11] or have named it pediatric inflammatory multisystem syndrome (PIMS) in several reports. [12] A French multicenter study reported on the disease, calling it ‘Kawa-COVID-19 [9]’. All reports mentioned above were published in early July. They emphasized that KD-like manifestations occurred in 30–70% of cases. However, because no cases of MIS-C or PIMS were observed during the early COVID-19 pandemic in Japan, we considered that the characteristics of MIS-C or PIMS would be different from those of original KD, which often develops in Japan.

In the present study, therefore, we evaluated the association between original KD cases at our hospital during the COVID-19 pandemic and SARS-CoV-2 infection using real-time reverse transcriptase-polymerase chain reaction (PCR) or loop-mediated isothermal amplification (LAMP) assay and/or qualitative tests for SARS-CoV-2 immunoglobulin G (IgG) and immunoglobulin M (IgM) by immunochromatography. Then we compared the characteristics of KD patients treated at our hospital between 1 April and 30 June with those during the same period in 2019 in our hospital, and those during the COVID-19 pandemic in Italy.

## 2. Materials and Methods

A retrospective, cohort, observational study was performed, and the clinical characteristics were compared for children with KD treated at our hospital from April to June 2020 and April to June 2019. Then we compared the characteristics of KD treated at our hospital from 1April 2020 to 31 August 2021 with the Italian series reported by Verdoni and colleagues [7].

Our pediatrics department is housed within a tertiary care institution, encountering around 1000–1500 inpatients per year, in Tokyo, Japan. The number of inpatients with KD was 69.0/year from 2012–2016. First, we compared the age, sex, percentage of insufficiency type, Kobayashi score value that predicts severity for KD, including age of onset, days of illness, percent of neutrophils, platelet count, and serum levels of aspartate aminotransferase, sodium, and C-reactive protein [13], and presence of coronary artery lesions between patients with KD treated at our hospital between April and June 2020 with those treated during the same time in 2019. Next, we evaluated the association between SARS-CoV-2 infection and KD patients at our hospital between April and June 2020, using PCR or LAMP assays for SARS-CoV-2 (Eiken Chemical Co., Ltd., Tokyo, Japan) and qualitative antibody tests for IgG and IgM (immunochromatography: INNOVITA^®^ 2019-nCoV Ab Test; Tangshan, China). These antibody tests were not able to be examined after July 2020 because of financial resources. Clinical characteristics, age of onset, Kobayashi score level, number of diagnosed cases of incomplete KD, blood test findings, concomitant symptoms, and proportion of patients presenting with KD shock syndrome (KDSS) and macrophage activation syndrome (MAS) were compared with the data of patients with KD during the COVID-19 epidemic in Italy. For the diagnostic criteria of KD, we used the AHA, 2017, for Italian cases [14]. For the study of our cases, we followed the revised fifth edition of the “Diagnostic Guidelines of Kawasaki disease” 2002 [15] to unify the condition of subjects in Japan. Coronary artery lesion is defined when the z-score of its inner diameter is +2.5 or more [16]. MAS is defined according to the classification in systemic juvenile idiopathic arthritis using a set of the Ferritin more than 684 ng/mL, and any two of platelet count ≤ 181 × 109/L, aspartate aminotransferase > 48 U/L, triglycerides > 156 mg/dL, and fibrinogen ≤ 360 mg/dL, which were also used for the case series in Italy [17]. KDSS is defined as KD patient with a sustained decrease in systolic blood pressure from baseline for age of 20% or more or with poor perfusion recognized clinically [18]. 

For the statistical test, Fisher’s exact test was used for the comparison of categorical variables, and the Mann–Whitney U test was used for comparison of continuous variables. We used JMP pro 14 (© SAS Institute Inc., Minato-ku, Tokyo, Japan) as a software for statistical program. This study was conducted with the approval of the Ethics Committee (Nihon University Research Committee #RK-200714-4, approved on 14 July 2020)**.**

## 3. Results

The profiles of KD patients in our hospital from April to June 2020 were compared to those from the same period in 2019. No significant difference was found in the number of patients, age of onset, sex, percentage of incomplete type, and Kobayashi score value. One case of coronary artery lesions in 2019 and two cases in 2020 were temporary and regressed to normal range within a month.

Then we listed and reviewed the characteristics of KD patients treated at our hospital between 1 April 2020 and 31 August 2021. All 32 patients were negative for PCR or LAMP for SARS-CoV-2 (0%, Table 1). All 18 patients who underwent qualitative testing for IgG and IgM were negative for both IgG and IgM (Table 1). One patient showed the symptoms for MAS. None of the patients developed KDSS. Only one patient presented with diarrhea as an accompanying symptom. One patient had convulsions and disturbance of consciousness as an accompanying symptom, but showed improvement.

The characteristics of KD patients during the COVID-19 epidemic in Italy are shown in Table 1 and Table 2 [7]. Because two patients were not detected either in PCR or antibodies’ testing, eight (80%) patients were confirmed SARS-CoV-2 infection. A PCR for SARS-CoV-2 from nasal swab was positive in two of the 10 cases. An antibody test was positive for IgG only in five cases, and both IgG and IgM were positive in three cases (Table 1). Diarrhea was seen as an accompanying symptom in six cases. MAS and KDSS developed in six and five patients, respectively.

Table 1 shows a summary of the results regarding the PCR and antibody tests for SARS-CoV-2 and treatments in our hospital and in Italy [7]. While 31% of the patients at our hospital required steroid treatment, 80% of the Italian children required adjunctive steroids, and 20% required inotropes. All items concerning SARS-CoV2 infection, steroid, and inotrope requirement were significantly different.

Table 2 shows comparisons of the clinical data of KD patients in our hospital and in Italy [7]. The age, high ratio of incomplete KD, Kobayashi score value and ratio of its high risk (≥5 points), CRP value, high ratio of neutrophils, low platelet count, low serum sodium, high ratio of MAS or KDSS, and high proportion of coronary artery dilation were significantly worse between the Italian group and our cohort. Among the parameters including Kobayashi score, all parameters, except for serum aspartate aminotransferase level in Italian cases, were significantly worse than those in ours. In the Italian group, the number of patients with incomplete KD was also high and many cases of MAS and KDSS were found.

## 4. Discussion

In 2020, the unexpected SARS-CoV-2 pandemic continued through summer and into the early autumn. We compared the clinical characteristics of KD patients during the same 3-month period with the previous year (2019) because the number of KD patients has a seasonal variation [4]. We found no significant difference of clinical characteristics in KD patients during the SARS-CoV-2 epidemic in 2020 compared to that in 2019, such as the age of onset and the severity of the disease using the Kobayashi score.

In Japan, a questionnaire survey polling major centers belonging to the Japanese Society of Kawasaki Disease showed no change in the morbidity and severity of KD. No case associated with COVID-19 infection was observed [18]. However, unfortunately, this survey did not investigate the detailed characteristics of the registered cases. As of this writing, one case, who developed KD after SARS-CoV-2 infection in Japan, was reported, but a causal relationship between the two diseases remained unknown [19]. Thereafter, despite the accumulation of pediatric cases with SARS-CoV-2 infections, no KD patients associated with COVID-19 appeared. We, therefore, conducted this current study to compare with representative cases associated with COVID-19 from Italy [7]. Compared between cases from Italy and our Japanese cases, the two cohorts were quite different in age of onset, Kobayashi score, and vasculitis markers, as well as a number of incomplete cases and accompanying symptoms, such as abdominal symptoms and neurological symptoms. In addition, comparisons with the reports from representative Western countries are summarized in Table 3 [6,7,8,9,10,11,12]. The onset age was high, including teenagers, and the race was predominantly African and Hispanic (Asian ethnicity in less than 5%). In addition, symptoms other than principal signs of typical KD, such as gastrointestinal and arthritic symptoms, hypotension, myocarditis, and MAS or KDSS symptoms, appeared in nearly half of the patients [6,7,8,9,10,11,12]. Intravenous immunoglobulin was used in almost all cases, but many were refractory [6,7,8,9,10,11,12].

In 2005, Esper et al. [20] conducted a study to determine the cause in eight out of 11 children with KD, who developed between November and April. They detected ‘New Heaven Corona Virus (HCoV-NH)’ by PCR in respiratory secretions from these KD patients, although only one was detected in 22 control cases. Thus, this virus was considered as one of the potential causes of KD. However, 47 among 48 patients in the United States and the Netherlands [21] and all 53 cases in Taiwan [22] during the same year had a negative result for HCoV-NH, resulting that HCoV-NH was not a cause of KD. Although HCoV-NH is an a-CoV genus by genetic classification, SARS-CoV-2 is a b-CoV genus, similar to the SARS virus from the greater horseshoe bat in the 2002 pandemic and the Middle East respiratory syndrome virus from the dromedary in the 2012 pandemic. The relationship between SARS-CoV-2 and KD would receive renewed attention in this era of the SARS-CoV-2 pandemic.

SARS-CoV-2 adheres to the angiotensin II-converting enzyme receptor upon invasion of infected target cells. [23] Then, increased inflammatory response, reactive oxygen release, vasoconstriction, and thrombus formation induce vasculitis and thrombosis. This response induces an activation of macrophages, which leads to neutrophil release and cytokines [24]. The markers of vasculitis, such as CRP, interleukin 6, D-dimer, and natriuretic peptide, are elevated in serum. As the results, the mucosal symptoms will appear like KD. At the same time, multisystemic vasculitis and subsequent tissue damage may develop a more severe form than typical KD [25]. The target organs are mostly pulmonary capillaries in adults, but in children, the gastrointestinal and cardiovascular systems seem to be more prevalent, leading to MIS-C or PIMS showing the symptoms similar to KD.

In Japan, a few cases of MIS-C or PIMS were reported after December 2020 [26,27,28,29]. We reviewed those reports according to the items in Table 1, and added Table 2, and their summary is added on Table 4. Comparing with Italy’s series and ours, Japanese MIS-C or PIMS cases are very close to Italy’s series rather than our KD series. Therefore, MIS-C or PIMS will possibly increase also in Japan after many more patients are infected with SARS-CoV-2. The majority of COVID-19 cases in Japanese children show very mild symptoms, such as low-grade fever or mild cough, or are asymptomatic [30]. However, only a few patients have a possibility to develop signs of MIS-C or PIMS-TS after several weeks in Japan.

Until October 2021, a few unpublished cases of MIS-C or PIMS were informed in addition to four published cases [26,27,28,29]. However, MIS-C or PIMS is still very rare, counting approximately 10 in Japan. As of this writing, the number of COVID-19 and MIS-C patients under 18 years old in the United States is approximately 5,900,000 and 5200, respectively [31,32], while those of Japan are approximately 260,000 [30] and around 10, respectively. We estimate that the morbidity of MIS-C or PIMS in Japan is markedly lower than that in the United States. The reason for such racial difference is expected to be a future study.

Considering the above discussions, it seems that the Kawasaki-like symptoms associated with SARS-CoV-2 infection may represent a different picture from the typical KD that we have in a single center in Japan. However, as described below, it is clear that SARS-CoV-2 is a vasculitis-causing virus, and coronary artery involvement was observed in 6–24% of the reported cases [26]. The inflammation of the coronary arteries may spill over into other arteries, as in KD. It is necessary to be cautious if there is an outbreak of KD signs in children during the COVID-19 pandemic.

As a limitation, our study, including the associated reports, was conducted at a specific time when the number of patients was small. It is necessary to accumulate more KD cases during the COVID-19 pandemic to elucidate the pathogenesis of MIS-C or PIMS.

## 5. Conclusions

Cases of COVID-19 epidemic-related KD in Italy show significantly different characteristics from the typical KD symptoms known in Japan. Although they meet the criteria for KD, we should consider them as a different type of vasculitis.

## Figures and Tables

**Table 1 children-08-00913-t001:** Summary of the results of the tests for severe acute respiratory syndrome coronavirus type 2 and treatments in our hospital and in Italy.

	Ours	Italy [7]	*p*-Value
Study period	April 2020–August 2021	March–April 2020	
Number of patients	32	10	
Nasal swab PCR for SARS-CoV-2	0/32 (0%)	2/10 (20%)	<0.05
Serology test for SARS-CoV-2 IgG	0/18 (0%)	5/10 (50%)	<0.01
Serology test for SARS-CoV-2 IgM	0/18(0%)	3/10 (30%)	<0.05
Use of adjunctive steroids	10/32 (31%)	8/10 (80%)	<0.01
Use of inotropes	0/32 (0%)	2/10 (20%)	<0.05

*p*-value was analyzed by Fisher’s exact test. PCR: real-time reverse transcriptase-polymerase chain reaction; SARS-CoV-2: severe acute respiratory syndrome coronavirus type 2; IgG: immunoglobulin G; IgM: immunoglobulin M.

**Table 2 children-08-00913-t002:** Comparisons of the clinical data of Kawasaki disease patients in our hospital and in Italy.

	Ours	Italy [7]	*p*-Value
Period of Investigation	April 2020–August 2021	March–April 2020	
Number of Patients	32	10	
Age (years)	1.8 ± 1.1	7.5 ± 3.5	<0.01
Male	21 (66%)	7 (70%)	0.8
Incomplete KD	1/32 (3%)	5/10 (50%)	<0.01
Kobayashi score	3.0 ± 2.1	5.1 ± 1.5	<0.01
Kobayashi score > 5	8/32 (25%)	7/10 (70%)	<0.01
CRP(mg/dL)	7.3 ± 5.0	25 ± 15.3	<0.01
Neutrophils (%)	62.2 ± 19.3	84.5 ± 5.7	<0.01
Platelets (×10^9^/L)	345 ± 122	130 ± 32	<0.01
Sodium (mEq/L)	135.7 ± 3.5	130.8 ± 3.9	<0.01
AST (U/L)	76.9 ± 101	87 ± 70	0.76
Fibrinogen (mg/dL)	526 ± 128	1176 ± 1032	0.14
MAS	1/32 (3%)	5/10 (50%)	<0.01
KDSS	0/32 (0%)	5/10 (50%)	<0.01
Coronary artery dilation	2/32 (6%)	6/10 (60%)	<0.01

*p*-value was analyzed by Fisher’s exact test or Mann–Whitney’s U test. KD: Kawasaki diseases, CRP: C-reactive protein, AST: aspartate aminotransferase, MAS: macrophage activation syndrome, KDSS: Kawasaki disease shock syndrome.

**Table 3 children-08-00913-t003:** The onset age and race of Kawasaki disease induced by severe acute respiratory syndrome coronavirus type 2 infections in Western countries.

Author & Area	Period	Number of KD-Like Pts.	Age	Sex (M:F)	Race
Verdoni et al. [7] Italy	17 March to 14 April	10	avg. 7.5	7:3	White 8/10
Riphagen et al. [6] UK	10 days in mid-April	8	4–14	5:3	A-C 6, M-E 1 Asian 1
Whittaker et al. [12] UK	23 March to 16 May	13	median = 9	10:3	Black 8, White 4 M-E 1
Toubiana et al. [8] France	27 April to 11 May	21	avg. 7.9	9:12	Black 12 (57%), Asian 3 (14%)
Pouletty et al. [9] France	6 weeks from April	16	median 10	8:8	A-C 10 (62%), M-E 2 (12%), European 4 (25%) Asian 0
Feldstein et al. [10] USA	15 March to 20 May	74	cKD Av. 5.7 iKD Av. 8.4	-	not written for KD patients
Dufort et al. [11] USA	before 10 May	36	0–5 y: 15/36 6–12 y: 18/36 13–20 y: 3/36	-	White 37%, Black 40%, Asian 5%, American native 18%

Av: average, KD: Kawasaki disease, y: years, cKD: complete KD, iKD: incomplete KD, A–C: Afro-Caribbean, M–E: Middle Eastern.

**Table 4 children-08-00913-t004:** Comparison with Japanese reported cases of MIS-C or PIMS.

	Mean	Uchida et al. [26]	Baba et al. [27]	Fukuda et al. [28]	Takasago et al. [29]
Number of Patients		1	1	1	1
Age (years)	11	16	10	9	9
Male	3/4 (75%)	M	F	M	M
Incomplete KD	1/4 (25%)	Y (3/6)	N (6/6)	N (6/6)	N (6/6)
Kobayashi score	5	6	4	5	5
Kobayashi score > 5	3/4 (75%)	Y	N	Y	Y
CRP(mg/dL)	22.1	20.6	21.9	22.6	23.1
Neutrophils (%)	92%	93%	91%	92%	90%
Platelets (x10^9^/L)	169	130	72	315	158
Sodium (mEq/L)	130	132	131	129	126
AST (U/L)	47	40	41	45	62
Fibrinogen (mg/dL)	665	606	605	758	691
MAS	0	N	N	N	N
KDSS	3/4 (75%)	Y	N	Y	Y
Coronary artery dilation	1/4 (25%)	N	Z = 3.88 (LMT)	N	N
duration after contact with SARS-CoV-2	28	23	unknown	30	31

## Data Availability

The data that support the findings of this study are available from the corresponding author (M.A.), upon reasonable request.

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
