# Peer review of "Characteristics of Kawasaki Disease Patients during the COVID-19 Pandemic in Japan: A Single-Center, Observational Study"

_children, 2021, doi:10.3390/children8100913_

Round 1
Reviewer 1 Report
This study compared clinical characteristics between KD patients under the period of COVID-19 pandemic in Japan and KD-like patients under the period of COVID-19 pandemic in Italy, showing that KD-like symptoms associated with SARS-CoV-2 infection may represent a different picture from the typical KD. This topic is of interest to the readers since the etiology of KD is still unknown. However, for the benefit of the readers, there are some points that need to be clarified:
- The aim of the study should be clearly stated both in the background of the abstract and at the end of introduction section.
- There have been many studies on COVID-19 and Kawasaki disease. What is the difference between this study and previous studies? What is the innovation? These need to be described in the introduction section.
- The definitions of incomplete KD and coronary artery lesions are not described in the methods section. Since the diagnostic criteria for KD is different for Japanese patients and Italian patients, are there any differences in the definition of incomplete KD and coronary artery lesions between the two cohort patients. Besides, the diagnostic criteria of MAS should be clearly stated.
- In the results section, no significant difference was found in the number of patients, age of onset, sex, percentage of incomplete type, and Kobayashi score value between KD patients in Japan from April to June, 2020 and those from the same period in 2019. What about the proportion of MAS and KDSS?
5. The English language needs to be improved.
Author Response
To Reviewer 1,
Thank you very much for your several important points to be revised.
Please refer to the attached file to reply.

Reviewer 2 Report
Although sample sizes were small, authors performed an informative study that would help understand and differentiate the pathogenesis of Kawasaki disease and multisystem inflammatory syndrome associated with SARS-CoV-2 infection. Thank you.
Author Response
To Reviewer 2,
We appreciate your kind review of our manuscript.
Please check our reply.

Reviewer 3 Report
This manuscript by Shimizu et al. render their observation to unravel the nature of characteristics of Kawasaki disease (KD) patients during the COVID-19 pandemic in Japan in a single center observational study. Nevertheless, a couple of issues should be detailed. First, few case of COVID-19 in Japan between April to June 2020. Second, exclude the result of no documented case of COVID-19 infection in their KD patients, we do not see any novel of interesting finding in this study. I encourage the authors show the results of 2021 and it will be more fit the title of this study.
Author Response
To Reviewer 3,
We appreciate your encouraging comment.
Please check our reply.

Round 2
Reviewer 1 Report
It would be better if the authors do a brief overview of the incidence and manifestations of covid-19 in Japanese children during this study.
Author Response
Reviewer 1
Comments and Suggestions for Authors
It would be better if the authors do a brief overview of the incidence and manifestations of covid-19 in Japanese children during this study.
Reply to Reviewer 1
Thank you very much for your advice to add a brief overview of covid-19 in Japanese children during this study. Concerning the manifestation of COVID-19 in Japanese, the severity is very low shown as on the website of the Ministry of Health, Labour and Welfare.
The incidence of COVID-19 in Japanese children is an important factor for our paper because it has been reported that the prevalence of MIS-C in Asian people seems lower than in other races. And we have shown the data of Japanese patient numbers summarized by the Ministry of Health, Labour and Welfare, and additionally, written about the markedly lower incidence of MIS-C. We revised the manuscript shown as below (in lines 215-225).
We hope this description is thought to be sufficient.
Additionally, we needed to inform you that we deleted Tables 1,2-1,2-2 because those can be summarized in Tables 3, 4 of the original manuscript according to the advice from another Reviewer. Then, the number of Tables 3, 4, 5, and 6 are changed to Tables 1, 2. 3, and 4, respectively.
Your understanding will be appreciated.
The majority of COVID-19 cases in Japanese children show very mild symptom such as low-grade fever or mild cough, or asymptomatic [31]. But only a few patients have a possibility to develop signs of MIS-C or PIMS-TS after several weeks in Japan.
Until October 2021, a few unpublished cases of MIS-C or PIMS are informed in addition to four published cases [27-30]. However, MIS-C or PIMS is still very rare to count approximately 10 in Japan. As of this writing, the number of COVID-19 and MIS-C under 18 years old in the United States is approximately 5,900,000 and 5,200, respectively [32, 33], while those of Japan is approximately 260,000[31] and around 10, respectively. We estimate that the morbidity of MIS-C or PIMS in Japan is markedly lower than that in the United States. The reason of such racial difference is expected to study from now.

Reviewer 3 Report
Thanks to authors' revision. However, the table 1, 2-1, and 2-2 can be remove from manuscript or they should add the characteristics of KD patients in 2021. Besides, the authors should add the description that how many percentages of COVID-19 test (PCR or antigen) in their KD patients in this pandemic.
Author Response
Reviewer 3
Comments and Suggestions for Authors
Thanks to authors' revision. However, the table 1, 2-1, and 2-2 can be remove from manuscript or they should add the characteristics of KD patients in 2021. Besides, the authors should add the description that how many percentages of COVID-19 test (PCR or antigen) in their KD patients in this pandemic.
Reply to Reviewer 3
Thank you very much for your second review of our manuscript. We agree with your direction to remove Tables 1, 2-2, 2-2 because they can be summarized into Tables 3, 4 of the original manuscript. We deleted them and the number of Tables 3, 4, 5, and 6 are changed to Table 1, 2. 3, and 4, respectively.
Then we wrote that ‘All 32 patients were negative for PCR or LAMP for SARS-CoV-2 (0%, Table 1).’ In lines 113-4 as the result of the COVID-19 test (PCR or antigen)
Additionally, we needed to inform you that we have added a brief overview of covid-19 in Japanese children during this study according to the comment by another reviewer. (in lines 215-225).
Your understanding will be appreciated.
The majority of COVID-19 cases in Japanese children show very mild symptom such as low-grade fever or mild cough, or asymptomatic [31]. But only a few patients have a possibility to develop signs of MIS-C or PIMS-TS after several weeks in Japan.
Until October 2021, a few unpublished cases of MIS-C or PIMS are informed in addition to four published cases [27-30]. However, MIS-C or PIMS is still very rare to count approximately 10 in Japan. As of this writing, the number of COVID-19 and MIS-C under 18 years old in the United States is approximately 5,900,000 and 5,200, respectively [32, 33], while those of Japan is approximately 260,000[31] and around 10, respectively. We estimate that the morbidity of MIS-C or PIMS in Japan is markedly lower than that in the United States. The reason of such racial difference is expected to study from now.
